# Attention-Based Multiscale Feature Pyramid Network for Corn Pest Detection under Wild Environment

**DOI:** 10.3390/insects13110978

**Published:** 2022-10-25

**Authors:** Chenrui Kang, Lin Jiao, Rujing Wang, Zhigui Liu, Jianming Du, Haiying Hu

**Affiliations:** 1School of Information Engineering, Southwest University of Science and Technology, Mianyang 621010, China; 2Institute of Intelligent Machines, Hefei Institutes of Physical Science, Chinese Academy of Sciences, Hefei 230031, China; 3School of Internet, Anhui University, Hefei 230601, China

**Keywords:** corn pest, convolution neural network, detection, attention, feature pyramid network

## Abstract

**Simple Summary:**

Corn pest recognition and detection is an important step for Integrated Pest Management. Generally, traditional methods adopt manual observation and counting in wild field to monitor the occurrence degree of corn pests. However, this is time-consuming and labor-intensive. An accurate and automatic corn pest detection method based on a deep convolutional neural network has been proposed in this paper. Extensive experimental results on a large-scale corn pest dataset show that the proposed method has good performance and can achieve precise recognition and detection of corn pests.

**Abstract:**

A serious outbreak of agricultural pests results in a great loss of corn production. Therefore, accurate and robust corn pest detection is important during the early warning, which can achieve the prevention of the damage caused by corn pests. To obtain an accurate detection of corn pests, a new method based on a convolutional neural network is introduced in this paper. Firstly, a large-scale corn pest dataset has been constructed which includes 7741 corn pest images with 10 classes. Secondly, a deep residual network with deformable convolution has been introduced to obtain the features of the corn pest images. To address the detection task of multi-scale corn pests, an attention-based multi-scale feature pyramid network has been developed. Finally, we combined the proposed modules with a two-stage detector into a single network, which achieves the identification and localization of corn pests in an image. Experimental results on the corn pest dataset demonstrate that the proposed method has good performance compared with other methods. Specifically, the proposed method achieves 70.1% mean Average Precision (mAP) and 74.3% Recall at the speed of 17.0 frames per second (FPS), which balances the accuracy and efficiency.

## 1. Introduction

Corn is a very important food crop in many countries. However, corn growth suffers from pests, resulting in great losses in corn production. Therefore, the early warning and forecast are the basis of the effective prevention and control of corn pests, which play important roles in agricultural management and decision-making for corn production. Until now, corn pests were detected by using the experience of agricultural technicians, which is time-consuming and uses up material resources. Additionally, the acquired information on corn pests is lagging in some remote areas. Thus, it needs to explore a fast, automatic, less expensive, and accurate method to address the detection of corn pests, which has great realistic significance.

In recent years, many machine-learning-based methods have been used for pest detection. For example, a scale-invariant feature transform (SIFT)-based feature learning method is used to recognize stone fly larvae images [1]; experimental results show that the recognition accuracy can achieve 82% for four classes of the stonefly. Based on the K-means cluster and correspondence filter, Faithpraise et al. developed a plant pest recognition system [2]. Wen et al. designed an image-based orchard insect automated identification and classification method using affine invariant local features, which achieves a classification rate of 86.6% [3]. Xie et al. proposed an automatic classification of field crop insects using multiple-task sparse representation and multiple-kernel learning [4]. However, the above machine-learning pest recognition methods need complex hand-crafted feature descriptors. Extracting accurate pest characteristics becomes more difficult when the backgrounds are complex in the wild environment.

Recently, because the convolutional neural network (CNN) has a powerful ability of feature extraction, the deep learning-based method has been widely applied to address the tasks of recognition and detection, which obtains great success. These object detection methods can be divided into proposal-based and proposal-free methods. For proposal-based methods, it can be divided into two parts, including region proposal generation and multi-classes detection. For example, Ross et al. proposed a region with CNN features (R-CNN) object detection method. It first generates about 2k region proposals for the input images, extracts a feature vector from each region proposal using CNN, and finally recognizes the class of each region proposal [5]. However, the detection efficiency is slow. Ross designed a fast region-based convolutional neural network method (Fast R-CNN) [6], which can improve the detection efficiency by using CNN to classify the object regions. However, the generation of object proposal is time-consuming. Ren et al. introduced Faster R-CNN [7] modules, which adopts a region proposal network (RPN) to produce proposals with few computations. To solve the multi-scale object detection problem, Lin et al. developed a feature pyramid network (FPN) [8], which can use multiple feature maps with different scales to recognize and localize objects. The feature maps of high level can be used to detect large objects and the feature maps of bottom layers with detailed information can be applied to detect small objects. To further improve the accuracy of object detection, multi-stage detectors are designed; for example, Cascade R-CNN detector [9]. Based on the two-stage detector, it adopts a three-stage detection method for multi-classes recognition and localization. However, the proposal-based methods will result in low efficiency. Thus, many region-free object detection methods have been proposed. These methods directly predict the localization and classes of objects. The traditional region-free detectors are the series of YOLO, including from YOLO v1 to YOLO v5 [10,11,12]. Different from YOLO, CornerNet predicts the top-left and right-bottom corners of the bounding box to achieve object detection [13]. FCOS adopts a new point-based prediction method, which further improves the detection speed and accuracy [14]. Based on the idea of points detection, a large number of point-based region-free detectors are proposed, including CenterNet [15], CentriptalNet [16], ExtremeNet [17], and so on, which perform well with object detection task.

Therefore, many researchers introduced CNN-based detection methods into agricultural pest detection. For example, a deep, learning-based, automatic, multi-class, wild pest recognition and localization method has been proposed using deep, hybrid, global, ad local activated features [18]. It could achieve 75.03% mAP on the built large-scale pest dataset, which outweighs other state-of-the-art methods. Wang et al. [19] have developed a deep convolutional neural network-based module to recognize pest images with 20 categories, which have a good practical significance for the intelligent identification of agriculture and forestry pests. Rahman et al. adopted state-of-the-art large-scale architectures such as VGG16 and InceptionV3 and fine-tuned them for detecting and recognizing rice diseases and pests [20]. Experimental results show the effectiveness of these models with real datasets. To address the precise detection of multi-class pests with small sizes, Jiao et al. [21] proposed an anchor-free, region-based, convolutional neural network (AF-RCNN) inspired by the working mechanism of the population visual system. Several experiments show that this method can obtain 56.4% mAP and 85.1% mRecall on a 24-classes pest dataset, which outperforms state-of-the-art methods at that time. Dong et al. introduced a multi-scale feature fusion network to detect multiply categories of pests, which has great improvements compared with other methods [22]. Additionally, to address the detection of aphids with tiny size and dense distribution, Teng et al. applied a transformer feature pyramid network and multi-resolution training method, which outperforms other state-of-the-art detectors [23]. Apart from the above pest detection methods, other CNN-based detection methods [24,25,26] are almost based on a two-stage object detector, Faster RCNN [7] and its modified versions [27,28], to identify and detect pests.

However, as we know that recent research rarely pays attention to the identification of corn pests; even if there are, only a small number of categories of corn pests are identified and detected. Thus, the lack of the corn pest dataset will hinder the precise recognition. Additionally, due to the influence of uncertain factors such as illumination and occlusion, the accuracy of current methods for corn pests with complex backgrounds is not high. Therefore, in this paper, we first built a large-scale corn pest dataset, including 7392 images with 10 types of corn pests. Then, to extract rich feature information from the corn pest image, we introduced deformable convolution into the deep residual network. Additionally, an attention-based, multi-scale, feature pyramid network is simultaneously designed, achieving accurate detection of corn with various scales. Finally, the proposed methods are combined with a Faster R-CNN detector into a unified network and achieve the detection of multiple classes of corn pests. The main contributions are listed as follows:(1)A deep residual network with deformable convolution is introduced to extract rich feature information of corn pests, which improves the expression ability of information of the network.(2)An attention-based multi-scale feature pyramid network is used to address the detection of corn pests of different sizes.(3)We have constructed a large-scale corn pest dataset, including 7392 corn pest images and 10 types of corn pests. By combining our method with the two-stage detector, the proposed method can achieve 70.1%mAP and 74.3% recall on the corn pest dataset.

## 2. Materials and Methods

### 2.1. Materials

#### 2.1.1. Corn Pest Image Collection

The corn pest images were collected in the wild using a mobile phone and camera from 2018 to 2020 in the Anhui and Henan provinces. All images were saved as JPG format with various sizes. The dataset contains 7392 images with 10 common types of corn pests, as shown in Table 1. It shows the names of these corn pests, the number of corn pest instances, and the average relative scale size to the whole pest images. Additionally, Figure 1 shows examples of each corn pest.

#### 2.1.2. Data Labeling

Data labeling is an important task in CNN-based object detectors. LabelImg software was used to annotate the label and location of each pest instance in an image by the professional plant protection expert. A pest in an image is usually labeled as (x,y,w,h,κ), where (x,y) represents the coordinate of the center of bounding box, (w,h) denotes the width and height of the bounding box, and κ is the class of the corn pest. Pest location coordinates and classes are saved as an XML file. The number of annotated samples corresponds to the number of bounding boxes labeled in each image. Every image could contain more than one annotation depending on the number and classes of pests.

#### 2.1.3. Data Splitting

To train and evaluate the performance of the CNN-based objector, all images and the corresponding annotations are randomly separated into the training set and testing set by a ratio of 9:1, including 6653 images for network training and 740 images for network testing.

#### 2.1.4. Analysis of Corn Pest Dataset

As shown in the left of Figure 2, we can observe that the relative size of the corn pest tends to be small, which will bring a great challenge to accurate detection. Additionally, the distribution of the number of pest instances is imbalanced which hinders the recognition of corn pests with few samples, as shown in the right of Figure 2.

### 2.2. Methods

In this section, we report implementation details of our proposed detector, as shown in Figure 3. First, we introduced the deep residual network with deformable convolution. We revisit the network architecture of the feature pyramid network (FPN) and analyze its working principle. Then, attention-based multi-scale feature fusion pyramid networks (AMFFP-Net) were developed based on FPN. Finally, we merged the proposed modules with the Faster R-CNN detector, achieving the recognition and localization of multi-classes corn pests.

#### 2.2.1. Deep Residual Network with Deformable Convolution Block

Deep residual network has a great ability to extract features, which performs well in various object detection tasks. However, for our corn pest detection task, there exist geometric transformations in pose, viewpoints, and part deformation of corn pest images, which bring great challenges for precise recognition and detection. Therefore, we introduce deformable convolution into deep residual network [29] which enhances the robust representation of corn pest.

The deformable convolution added an offset based on regular convolution; the parameters of the offset can be obtained by learning. Compared with regular convolution, the sampling space of deformable convolution is enlarged by adding the offsets; therefore, the area of receptive field is changed. To be specific, the process of deformable convolution can be defined by Equation (1):(1)$$y(l_{0})=\sum_{p_{n}\in \Omega }w(l_{n})\cdot x(l_{0}+l_{n}+\Delta l_{n})$$
where $y(l_{0})$ represents the output of each location *l*_0_ in input x; Ω represents the sampling space in the input feature map x; w is the learnable weight; $l_{n}$ enumerates the location of sampling space Ω. $\Delta l_{n}$ denotes the learnable offset, which can be obtained by the convolutional neural network.

Since the position after adding the offset is not an integer and does not correspond to the actual pixel points on the feature map, it is necessary to use interpolation to obtain the offset pixel values. Usually, the bilinear interpolation can be used. The formula is as follows:(2)$$\begin{array}{l} x(l)=\sum_{k}G(k,l)\cdot x(k)\\ =\sum_{k}g(k_{x},l_{x})\cdot g(k_{y},l_{y})\cdot x(k)\\ =\sum_{k}\max(0,1-\left|k_{x}-\left.l_{x}\right|\right.)\cdot \max(0,\left|k_{y}-\left.l_{y}\right|\right.)\cdot x(k) \end{array}$$
where G(k,l) denotes the two dimensional bilinear interpolation kernel and l represents the arbitrary location ($l=l_{0}+l_{n}+\Delta l_{n}$).

From Figure 3b, we can observe that the offset can be learned from a convolutional layer, and the kernel size is 3×3 with Dilation 1. Therefore, the size of the output feature map is the same with the input feature map. When training the network, we use a back-propagated (BP) algorithm to learn the parameter of the offsets.

#### 2.2.2. Attention-Based Multi-Scale Feature Fusion Pyramid Network (AMFFP-Net)

To obtain the multi-scale features of corn pest images, we learn from feature pyramid network (FPN) [8], which merges the low-level and high-level feature maps. In this paper, we proposed an attention-based multi-scale feature fusion pyramid network to enhance the expression ability of FPN. In this section, we first describe the architecture of the FPN network, then introduce our proposed attention-based, multi-scale, feature fusion pyramid network in detail.


**(1) Revisiting feature fusion in FPN**


Two key elements, including the downsampling factor and the fusion proportion between adjacent layers, affect the performance of FPN. Previous works improve the performance by decreasing the downsampling factor; however, this will lead to an increase in the computation complexity.

In this section, we provide the background of the FPN [8]. Let B denote the 1×1 convolutional operation for changing channels, and $F_{up}$ denotes upsampling operation for increasing solutions. Therefore, the aggregation of adjacent feature layers in the following manner:(3)$$P_{i}=B_{i}(X_{i})+\alpha F_{up}(P_{i+1})$$
where, α represents the fusion factor between two different adjacent layers, which is set to 1 in FPN.


**(2) Attention-based feature fusion**


We can observe that the fusion factor in FPN is the same regardless of the layers of feature maps. This will result in poor distinguishability during feature fusion between different layers. Therefore, in this study, we have added a learnable fusion factor to increase the distinguishability, which can benefit the recognition of different objects. Figure 3a shows the network architecture of our attentional FPN. In this figure, we only take three layers of a pyramid network as an example; however, we adopted feature maps from five residual blocks of ResNet [29] in the proposed AMFFP-Net module. Similar to FPN, all feature maps generated by each residual block were processed by a 1×1 convolutional layer for reducing the number of channels. Specifically, the feature map F_3_ was 2× up-sampled by nearest interpolation and was thens fed into an attentional weight generator for producing the weights used in feature fusion. Then, the feature map F3 with attentional weights was fused with feature map F2. It is noted that, similar to FPN, our AMFFP-Net has five outputs; the top features P5 and P6 can be obtained by twice subsampling. Finally, we append a 3×3 convolutional layer to eliminate the aliasing effect.

The fusion process also can be represented as Equation (3). Different from FPN, the α in our AMFFP-Net is changeable. Feature maps from different levels have different $\alpha_{s}$. Thus, there are different $\alpha_{s}$ in our AMFFP-Net module. In AMFFP-Net, the $\alpha_{s}$ are developed by the attentional weights generator, as shown in Figure 3c. The attentional weights generator consists of a convolutional layer with a 1×1 kernel size, a ReLu activation function for non-linear transformation, a convolutional layer with 3×3 kernel size, and a sigmoid function used for generating weight maps.

#### 2.2.3. Joint Detection

Following a two-stage detector, e.g., Faster R-CNN [7], we combined the proposed deep residual network with deformable convolution and AMFFP-Net with Fast R-CNN [6] detector. To be specific, a region proposal network (RPN) has been used to generate a set of pest proposals. Then, these corn pest proposals are input into the two fully connected layers, followed by the classification layer with (*c* + 1) outputs (*c* is the number of classes of corn pests) and the localization layer with 4*c* outputs. Finally, we obtained the name of the corn pest and its corresponding location in an image.

### 2.3. Evaluation Metrics

To evaluate the detection accuracy of our model and other compared methods, several standard metrics are applied, such as mean Average Precision (*mAP*) and *Recall*, which can be calculated as follows. First, the *IoU* is used to verify the overlap between the ground-truth and predicted bounding box, which can be defined:(4)$$IoU=\frac{area(G\cap P)}{area(G\cup P)}$$
where *G* and *P* denote the ground truth and predicted bounding box, respectively. The area(G∩P) denotes the intersection of the ground truth and predicted bounding box and area(G∪P) denotes the union of the ground truth and predicted bounding box.

Secondly, according to the value of *IoU*, we decided the true positive (*TP*) and false positive (*FP*). If the *IoU* of the predicted bounding box and ground truth is greater than 0.5, then the predicted bounding box is viewed as *TP*. Otherwise, it is *FP*. Thus, the *precision* and *recall* can be calculated using:(5)$$precision=\frac{\#TP}{\#TP+\#FP}$$
(6)$$recall=\frac{\#TP}{GT}$$
where #TP and #FP represent the number of detected and misdetected corn pests, respectively. Ground Truth (*GT*) denotes the total number of corn pests.

The precision and recall metrics are combined to fairly evaluate the performance of our method. Thus, the Average Precision (*AP*) was adopted to verify the models. The *AP* denotes the area under the precision/recall curve, which can be defined in Equation (7). Mean AP (*mAP*) averaged over all object classes is employed as the final measure to compare performance on all object classes, and it is defined as Equation (8):(7)$$AP=\int_{0}^{1}PdR$$
(8)$$mAP=\frac{1}{c}\sum_{j=1}^{c}AP_{j}$$
where *c* is the number of classes, which is set to 10 in this work.

## 3. Results and Analysis

### 3.1. Experimental Platform and Parameters Setting

**Experiment platform:** All experiments of this work were run on a workstation equipped with one NVIDIA RTX 2080Ti GPU with 24 GB memory. The software environment is Ubuntu 18.02 from Canonical Ltd in London, UK, which is an open source software, and Python 3.8 designed by Guido van Rossum in Commonwealth of Virginia, American. All CNN-based models have been built using Pytorch, which is an open source framework designed by Facebook.

**Parameters setting**: The parameters of each comparison detection framework used in this paper are consistent with their default parameters without any adjustment.

### 3.2. Experimental Results and Analysis

Detection results of the proposed method and compared method are shown in Table 2. We can observe that our method can achieve 74.3% mean *Recall* and 70.1% mean *AP*, which obtains the improvements of 4.9%, 0.5%, and 2.8% *AP* compared with FPN [8], S-RPN [26], and Cascade R-CNN [9], respectively. This demonstrates that the deep residual network with deformable and attention-based multi-scale feature pyramid network contributes to the gain of the performance.

Additionally, we further explored the detection accuracy of each category of corn pest. In Table 2, we can see that the precision of class “DP” only has 44.0% AP and 46.3% recall, which is lower than other classes of corn pest. We found that serious occlusion will significantly decrease the recognition accuracy, as shown in Figure 4. In the next work, we will focus on this challenge.

### 3.3. Detection Efficiency

The detection efficiency is a metric that needs to be taken into consideration in the real application. In this paper, we use three different metrics to verify the efficiency of the proposed method. The result is reported in Table 3. It shows that our method can achieve 17.0 FPS, leading to the improvements of 2.5 and 3.9 FPS compared to S-RPN and Cascade R-CNN, respectively. However, its speed is slightly lower that of the FPN. From the view of GLOPs and the number of parameters, we can observe that it is slightly inferior to the FPN detector.

### 3.4. Ablation Experiment

We carried out a series of experiments to explore the effect of the deformable convolution and AMFFP-Net. The detection results on the corn pest dataset are reported in Table 4. Here, we take the Faster R-CNN detector with ResNet50 backbone as the baseline. When we introduced deformable convolution instead of conventional into a deep residual network, the *mAP* rose from 65.2% to 66.3%, implying that the deformable convolution has contributed to object detection. Additionally, when we adopted the proposed AMFFP-Net, the performance of our method could improve by 3.8% *mAP*, demonstrating that the attention mechanism is useful. From the evaluation metric of Recall, the trend is similar to that of *mAP*.

### 3.5. Visualized Detection Results and Analysis

To verify the effectiveness of our proposed method, we also visualized some detection results on the corn pest dataset, as shown in Figure 5. It demonstrates that the proposed method can achieve the accurate recognition and localization of corn pests. However, we also found that there exist some poor results; for example, some corn pests are undetected, as shown in Figure 6. On the left of this figure, the pest is not detected using our method because the pest instance is concealed in the background. This will hinder the precise detection of corn pests. Additionally, we also observed that the dense distribution of small corn pests also causes poor performance in the proposed detector, as shown in the middle and right figures of Figure 6. These problems need to be addressed in future work.

## 4. Conclusions and Future Work

Due to the complexity of the corn pest image, such as the complex background, the various size of corn pest instances, and so on, these problems bring great challenges for precise corn pest detection. In this paper, we first introduced deformable convolution into a deep residual network and then developed an attention-based multi-scale feature pyramid network for enhancing the feature representation of the corn pest image. Finally, these proposed modules were combined with a two-stage detector to achieve the detection of corn pests. Experimental results on large-scale corn pest datasets show that the detection results of our method outperform other approaches in terms of accuracy and speed. However, in Section 3.2, we see that dense distribution and serious overlapping of corn pests have great influence on the accuracy of detection. Therefore, in future work, we plan to adopt a coarse-to-fine strategy to address the problems of dense distribution and serious overlap.

## Figures and Tables

**Figure 1 insects-13-00978-f001:**
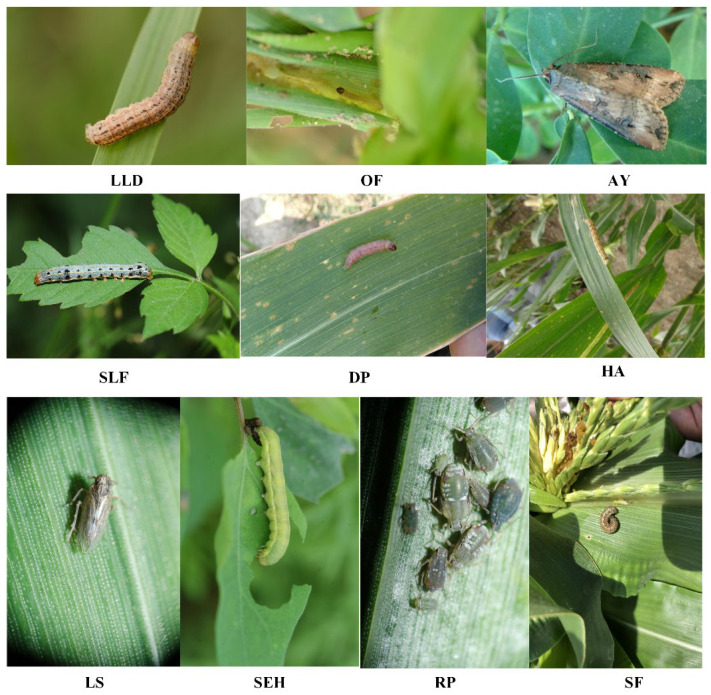
Examples of 10-types of corn pest.

**Figure 2 insects-13-00978-f002:**
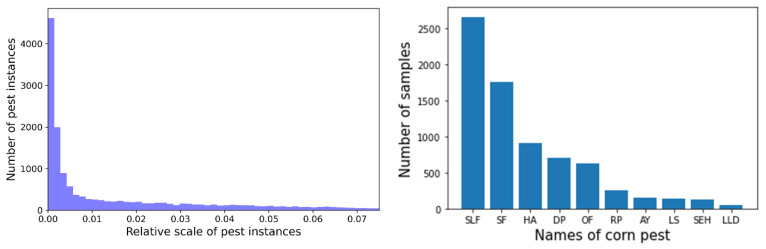
Distribution of relative scale of corn pest instances.

**Figure 3 insects-13-00978-f003:**
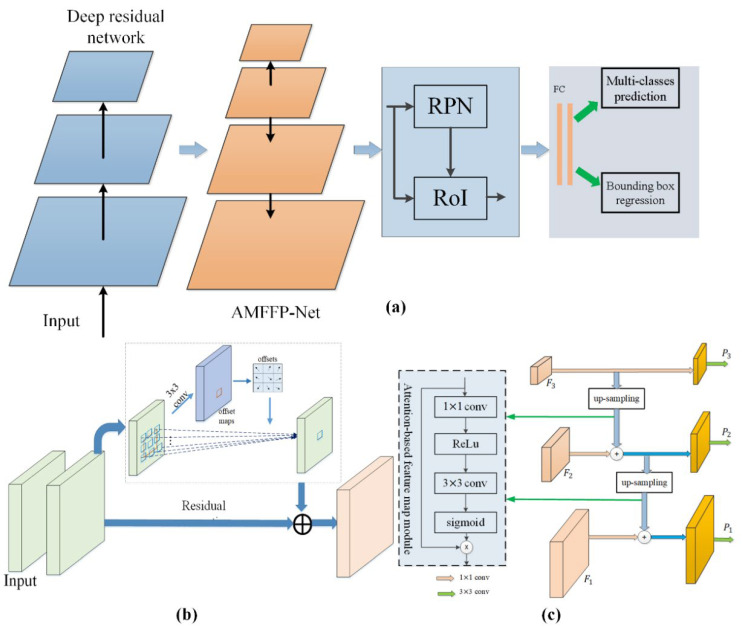
(**a**) Overview of the pipeline proposed corn pest detection module. (**b**) Deep residual network with deformable convolution block. (**c**) Architecture of the proposed aFPN. The weights generator is used to produce a set of attentional weights, which is related to the upper layers.

**Figure 4 insects-13-00978-f004:**
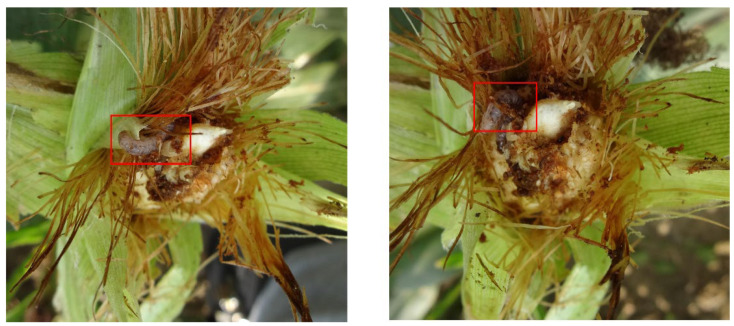
Examples of serious occlusion problem. The red boxes denote the location of the corn pest.

**Figure 5 insects-13-00978-f005:**
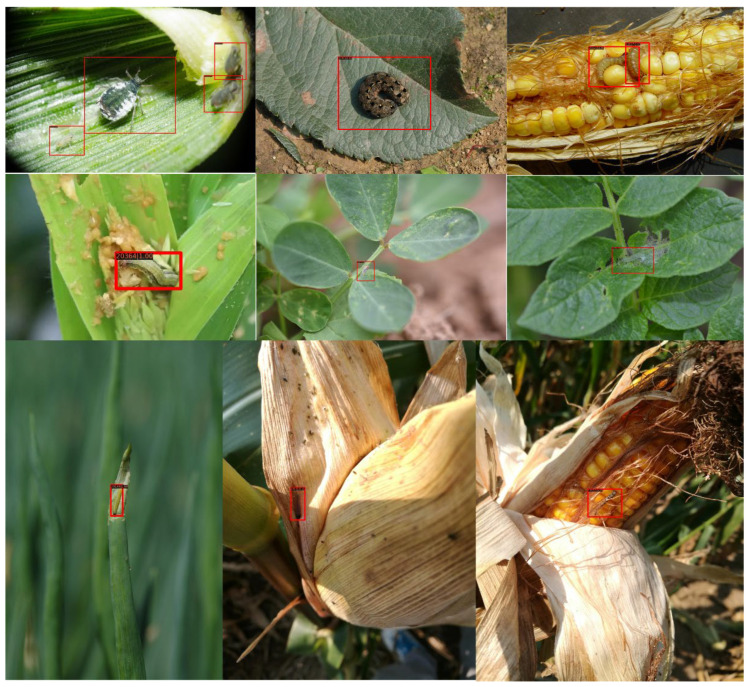
Some examples of visualized detection results of the proposed method. The red boxes represent the predicted location of corn pest.

**Figure 6 insects-13-00978-f006:**
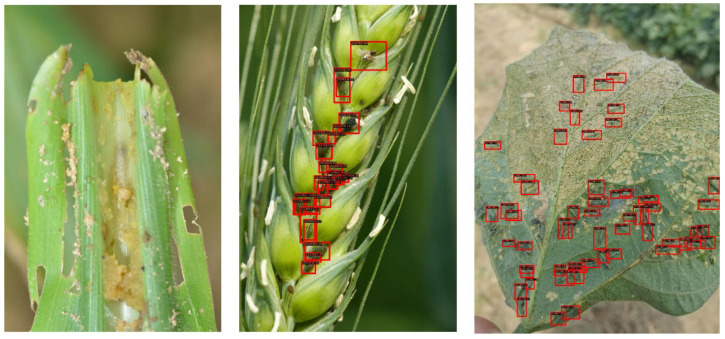
Poor detection results of the proposed method. The red boxes represent the predicted location of corn pest.

**Table 1 insects-13-00978-t001:** Description of corn pest dataset used in our work.

ID	Scientific Name	Number of Images	Number of Corn Pest Instances	Average Relative Size
1	*Leucania loreyi Duponchel* (LLD)	55	55	0.192
2	*Ostrinia furnacalis* (OF)	629	650	0.042
3	*Agrotis ypsilon* (AY)	146	174	0.153
4	*Spodoptera litura Fabricius* (SLF)	2664	7976	0.306
5	*Dichocrocis punctiferalis* (DP)	709	849	0.038
6	*Helicoverpa armigera* (HA)	916	919	0.094
7	*Laodelphax striatellus* (LS)	139	140	0.061
8	*Spodoptera exigua Hiibner* (SEH)	131	141	0.048
9	*Rhopalosiphum padi* (RP)	249	3875	0.007
10	*Spodoptera frugiperda* (SF)	1754	1970	0.057

**Table 2 insects-13-00978-t002:** Detection results of the proposed method and compared detectors on corn pest dataset.

Class	FPN		S-RPN		Cascade R-CNN		Our Method	
	*Recall*	*AP*	*Recall*	*AP*	*Recall*	*AP*	*Recall*	*AP*
LLD	83.3	81.8	100	100	100	100	100	100
OF	60.6	56.4	67.6	59.1	59.2	51.6	69.0	60.0
AY	88.2	74.5	83.1	80.1	88.9	81.8	83.3	81.8
SLF	56.0	49.7	63.4	58.3	55.4	49.6	64.1	58.1
DP	48.8	45.5	46.0	44.6	46.3	44.7	46.3	44.0
HA	85.9	79.4	80.4	79.4	81.5	78.5	82.6	79.6
LS	100	100	100	100	100	100	100	100
SEH	60.0	54.5	58.8	53.6	58.8	52.9	58.8	53.6
RP	61.1	48.9	68.4	58.5	62.0	51.4	70.1	61.7
SF	69.4	61.4	68.6	62.1	67.0	62.1	69.1	62.5
Mean	71.3	65.2	73.6	69.6	71.9	67.3	74.3	70.1

**Table 3 insects-13-00978-t003:** Detection efficiency of the proposed method and other compared methods on a single NVIDIA GPU.

Method	Speed (FPS)	GFLOPs	Number of Parameter (M)
FPN	18.2	216.34	41.17
S-RPN	14.5	241.12	46.23
Cascade R-CNN	13.1	244.13	68.95
Ours	17.0	224.22	41.82

**Table 4 insects-13-00978-t004:** Ablation experimental results based on the baseline (Faster R-CNN detector).

Deformable Convolution	AMFFP-Net	*mAP*	*Recall*
		65.2	71.3
√		66.3	69.5
√	√	70.1	74.3

## Data Availability

Not applicable.

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
