# Peer review of "Attention-Based Multiscale Feature Pyramid Network for Corn Pest Detection under Wild Environment"

_insects, 2022, doi:10.3390/insects13110978_

Round 1
Reviewer 1 Report
General comments:
Insect pests severely affect both corn production. To prevent damage caused by corn pests, the accurate and robust corn pest detection is an important step in the stage of early warning. To achieve accurate detection of maize pest, authors proposed an attention-based multiscale feature pyramid network. Experiment results show that the proposed method in this study can obtain an encouraging detection performance compared to those state-of-the-art algorithms. This work is meaningful in the field. Clearly, this research fits into the scope of the journal, and it could contribute towards furthering the current state of knowledge in this field, below are some of my major comments.
* strong points *
Much work is done in this paper, including:
(1) At present, a deep residual network with deformable convolution has been introduced to extract rich feature of corn pest under complex background.
(2) To address the detection task of multi-scale corn pest, authors developed an attention-based multi-scale feature pyramid network. And by combining the proposed modules with two-stage detector into a single network, achieving the identification and localization of corn pest in an image.
(3) Authors constructed a large-scale corn pest dataset, including 7741 images with 10types of pests, which provides an important data base for research in this field.
* three weak points *
(1) In section 2.1, it is common practice to indicate details of the data collection, such as species, time, location, etc., that need to be added.
(2) The language should be polihsed by native English speaker.
(3) Some latest efforts should be discussed in this work.
[1] Automatic Crop Pest Detection Oriented Multiscale Feature Fusion Approach [J]. Insects, 2022, 13, 554. https://doi.org/10.3390/insects13060554.
[2] TD-Det: A Tiny Size Dense Aphid Detection Network under In-Field Environment [J], Insects 2022, 13, 501. https://doi.org/10.3390/insects13060501
For these reasons I suggest to accept the paper after modifying the present form.
Author Response
Reviewer #1:
General comments:
Insect pests severely affect both corn production. To prevent damage caused by corn pests, the accurate and robust corn pest detection is an important step in the stage of early warning. To achieve accurate detection of maize pest, authors proposed an attention-based multiscale feature pyramid network. Experiment results show that the proposed method in this study can obtain an encouraging detection performance compared to those state-of-the-art algorithms. This work is meaningful in the field. Clearly, this research fits into the scope of the journal, and it could contribute towards furthering the current state of knowledge in this field, below are some of my major comments.
* strong points *
Much work is done in this paper, including:
(1) At present, a deep residual network with deformable convolution has been introduced to extract rich feature of corn pest under complex background.
(2) To address the detection task of multi-scale corn pest, authors developed an attention-based multi-scale feature pyramid network. And by combining the proposed modules with two-stage detector into a single network, achieving the identification and localization of corn pest in an image.
(3) Authors constructed a large-scale corn pest dataset, including 7741 images with 10 types of pests, which provides an important data base for research in this field.
Answer: Thank you very much for your affirmation and suggestions for our work. As your comments, we have revised and improved the manuscript carefully. More details are listed as follows:
* three weak points *
(1) In section 2.1, it is common practice to indicate details of the data collection, such as species, time, location, etc., that need to be added.
(2) The language should be polished by native English speaker.
(3) Some latest efforts should be discussed in this work.
[1] Automatic Crop Pest Detection Oriented Multiscale Feature Fusion Approach [J]. Insects, 2022, 13, 554. https://doi.org/10.3390/insects13060554.
[2] TD-Det: A Tiny Size Dense Aphid Detection Network under In-Field Environment [J], Insects 2022, 13, 501. https://doi.org/10.3390/insects13060501
For these reasons I suggest to accept the paper after modifying the present form
Question 1: In section 2.1, it is common practice to indicate details of the data collection, such as species, time, location, etc., that need to be added.
Answer 1: Thanks so much for your valuable comments, according to your suggestion, we have added details of the data collection, such as species, time, location.
Question 1: The language should be polished by native English speaker.
Answer 2: Thanks for your comments. We have carefully checked the grammar of our manuscript and polished it.
Question 3: Some latest efforts should be discussed in this work.
[1] Automatic Crop Pest Detection Oriented Multiscale Feature Fusion Approach [J]. Insects, 2022, 13, 554. https://doi.org/10.3390/insects13060554.
[2] TD-Det: A Tiny Size Dense Aphid Detection Network under In-Field Environment [J], Insects 2022, 13, 501. https://doi.org/10.3390/insects13060501
Answer 3: Thanks so much for your valuable comments, according to your suggestion, we have added some more in-depth literature research in the Introduction.
Finally, we tried our best to improve the manuscript and made some changes in the manuscript. And here we did not list the changes but marked in red in the revised paper. We appreciate for Editors’ and Reviewers’ warm work earnestly and hope that the correction will meet with approval. We look forward to hearing from you regarding our submission. We would be glad to respond to any further questions and comments that you may have.

Reviewer 2 Report
Dear authors,
congratulation on your excellent work. Some presentation details should be addressed. I added some comments directly to the manuscript.
Best regards

Author Response
Reviewer #1:
Comments:
congratulation on your excellent work. Some presentation details should be addressed. I added some comments directly to the manuscript
Answer: Thanks so much for your comments and suggestions. We have carefully study your comments and revised, and the revised parts are marked up using the “Track Changes” function.
Finally, we tried our best to improve the manuscript and made some changes in the manuscript. And here we did not list the changes but marked in red in the revised paper. We appreciate for Editors’ and Reviewers’ warm work earnestly and hope that the correction will meet with approval. We look forward to hearing from you regarding our submission. We would be glad to respond to any further questions and comments that you may have.
